# "The Melancholy Dames": Soren Kierkegaard's Despairing Women and Wesley's Empowering Cure

**Diane Leclerc** [1,2]

¹ Department of Religion, College of Theology and Christian Ministry, Nashville, TN 37212, USA; dkleclerc@nnu.edu
² Historical Theology, Northwest Nazarene University, Nampa, ID 83686, USA

**Abstract:** This article will bring together the work of Soren Kierkegaard and John Wesley for the purpose of showing the relevance of their theologies for the empowerment of women. The particular focus will be on the doctrine of original sin. The paper will first address the question of why Augustine's novel doctrine became the orthodox position and why his construction restricts its applicability to women. It will then move to Soren Kierkegaard's understanding of anxiety and despair in his treatise, *The Sickness Unto Death*. In the theology of Soren Kierkegaard, there is room to interpret his understanding of original sin as "gendered". For him, despair is the counterpart of original sin. It finds two forms: 1. despair is willing to be a self apart from the Power (God) that constitutes the self, and 2. despair is not willing to be a self at all. Feminists have questioned the legitimacy of original sin in its traditional form, and a few have even used Kierkegaard on the way to offering an alternative to pride. One method used here is to explicate this insight further. Another method is to put Kierkegaard and John Wesley in dialogue for the purpose of imagining selfhood for women more hopefully. If "despair" can be imagined as a wounding of the self, Wesley's therapeutic model—seeing original sin as a disease and sanctification as its cure—has much to offer the conversation on personhood and empowered subjectivity, particularly for women. The primary research question investigated here is how a conversation between feminism, Kierkegaard, and Wesley offers an alternative to Augustine's "orthodoxy" without rendering the idea of original sin completely untenable and useless for women within Christianity. Even though Wesley's curative paradigm has been highlighted in more recent years, its particular strength to speak into the lives of those who do not/cannot will to be a self has perhaps yet to be fully mined. It reveals itself in the entire Wesleyan history of affirming women. However, the author believes the potential power of Wesley's theology can be further unleashed by examining its mechanism's in countering "female despair".

**Keywords:** Soren Kierkegaard; John Wesley; gender; original sin

## 1. Introduction

It is certainly evident that some cultures have reached a place of identifying 'gender' as a malleable, even indeterminate category. Not only is the female "a sex which is not one", as Luce Irigaray posited, but gender has also become a dyad which "is not two". In light of the fact that the question can be asked, "whatever happened to sin", and in light of the fact that "female" has long been defined (quite inappropriately) only as the not-male (or more recently as an antiquated denotation for one gender among many), my aim to construct a theology that speaks meaningfully of "women's sin" could render me, in all practicality, quite speechless. What is feminism, theoretically, when, practically, fewer persons identify solely as female, and the category is viewed more tentatively? Has feminism's participation in gender theory and analysis landed it in a space where it has no real subject? In order to have something to say, I have taken what is known as a strategic essentialist position in relation to "woman's sin" and, in doing so, have suggested a hamartiology that has

addressed the theological significance of "women's experience". In other words, just as strategic essentialism attempts to confiscate female subjectivity back from male possession, my affirmation of "women's sin" has been a strategic attempt to challenge the singularity of sin as pride and to overturn that paradigm's negative effects on women themselves. It has been my conviction that defining original sin as pride disrupts the female subjective experience by exhorting women to remain humble, submissive, complicit, and silent under the guise of womanly virtue. It has also been my conviction that an alternative image of sin opens the possibility for a quite different signification of (female) holiness: shifting the definition of the "not-good" necessarily changes the meaning of the "good". The "holy woman" takes on a radically different connotation when images of sin are freed from (their) hubris.

This article is an extension of my work in *Singleness of Heart: Gender, Sin, and Holiness in Historical Perspective* (Leclerc 2001). It will offer a brief overview of the development of Augustine's doctrine of original sin, but its main purpose is to focus on the ideas of Soren Kierkegaard's understanding of sin as despair in conversation with John Wesley's soteriology as a means of offering a therapeutic cure for Kierkegaard's concept of the sickness unto death, especially for "despairing women, or "melancholy dames".

## 2. Discussion

### 2.1. Where It All Began

Scholars, who *do not* simply accept Augustine's doctrine of original sin as a wondrously providential gift to all subsequent reflection on theological anthropology, or as a just corrective to the first three centuries of Christian thought have attempted to explain the norming of Augustine's very novel and idiosyncratic understanding of sin by turning to themes such as sexuality, politics, and the Hellenistic worldview. Peter Brown interprets Augustine by reflecting on the power of sexuality in Augustine's life and, thus, in his theology. Elaine Pagels offers insight into the changing and shifting political and social situation that would have allowed Augustine's understanding of humanity to triumph. Additionally, Elizabeth Clark, by exploring the history of the Origenist controversy, contrasts the earlier, almost strident defense of human free will found in Origen and other orthodox theologians against Augustine's final defense of a Christianized determinism in his battle against Pelagius.

Each author is grappling with the same intriguing question: how could Augustine's doctrine of original sin come to be accepted as the orthodox position in light of all that came before it? Or, in Pagel's words, "Why did his teaching on original sin become the center of western Christian tradition, displacing, or at least wholly recasting, all previous views of creation and free-will?" (Pagels 1988). It requires at least some historical investigation to even perceive and recognize that for Augustine's doctrine to be accepted as orthodox, some sort of radical shift of context had to take place. Brown, Pagels, and Clark offer noble attempts at this historical, social, and theological reconstruction that place Augustine firmly in his context. Additionally, it is here, in his own context, that questions about the doctrine of original sin itself quite naturally arise, attempting to ask the question of the legitimacy of original sin while chasing the free-floating theological giant, as one would chase a balloon without a tether, is hardly productive in the theological task. Yet, how often has Augustine been allowed to float about at will? Context, offered by Brown, Pagels, and Clark, makes a "face to face" inquiry theoretically possible.

Peter Brown's study of the body, and his investigation of Augustine in particular, illuminates the place of the body as a religious symbol in antiquity. Brown places Augustine in a historical and ecclesiastical context in which the body could reveal "greater" truth about humanity or be denied as a means toward the greater perfection of the soul (Brown 1988). No doubt, Augustine's Platonist and Manicheaen influences shape his concepts of body and sexuality. However, it is also the ecclesiastical milieu of fourth-century asceticism that molds Augustine's understanding. Brown offers an important corrective in showing that Augustine does not go as far as Jerome or others in rejecting the marital relationship as

somehow evil. Yet, Augustine's ascetic commitments still wage war with his own "inner corrosion" of lust that he believes is almost hopelessly independent from his own will. Augustine universalizes his own struggle into an understanding of a damaged will passed down from Adam. In contrast, the ascetic Pelagius, who was apparently able to master his will as it related to his sexuality, and the married Julian reject Augustine's understanding of a corrupted will due to their very different life experiences. Different person's very different experiences of the "body" call Augustine's universalizing tendencies into question.

While sexuality is certainly not a silent issue in Pagel's interpretation of Augustine, her focus is "government", i.e., how did fourth-century Christian citizens view their relationship to secular political powers? Apparently, the belief that persons are fully capable of self-government pervaded Christianity prior to Augustine. "Yet, with Augustine, in the late fourth and early fifth centuries, the message is changed . . . Instead of freedom of the will and humanity's original royal dignity, Augustine emphasizes humanity's enslavement to sin" (Pagels 1988). While Chrysostom, for example, would declare that Christian baptism restores the capacity for self-government, Augustine's more pessimistic view states that a hierarchical government is wholly necessary; human beings, thoroughly corrupted, cannot be trusted and therefore an external, even secular means of law and justice is necessary. In fact, Augustine completely transforms the Christian meaning of self-government from a call and right proclaimed by the liberation inherent in the Christian message to a sign of sin itself, an almost incredulous shift! Pagels points out that Justin and Irenaeus affirm that Christians have recovered the damage inflicted by sin and that other patristics concur: "The desire to master one's will, far from expressing what Origen, Clement, and Chrysostom consider the true nature of rational beings, becomes for Augustine the fatal temptation . . . Augustine cannot resist reading the desire for self-government as total, obstinate perversity" (Pagels 1988).

Why would Christians accept this understanding of humanity and its extension to a particular political stance? Through a significant historical review, Pagels reveals that Augustine's concept of human corruption could explain the observable decline in morality and virtue in the Church since the empire was Christianized. "Augustine's theory of original sin could make theologically intelligible not only the state's imperfections but the church's imperfections as well" (Pagels 1988). Further, "It is Augustine's theology of the fall that made the uneasy alliance between Catholic churches and imperial power palatable—not only justifiable but necessary . . . " (Pagels 1988). Pagels argues that people were ready to hear a theological explanation for the changing and often confusing imperial and ecclesiastical situation.

Clark picks up the same general theme, as seen in Origen's argument for Christianity's belief in human freedom against gnostic determinism and then in Pelagius's defense of free will against Augustine's new "purification" of determinism that had long been seen as anti-Christian. (Clark 1992). Clark, similar to Pagels and Brown, attempts to make sense of this great historical irony. Clark traces the development from Origen to Pelagius by reviewing the contributions of some "post-origenists" and "pre-pelagians". While Clark appropriately identifies all the theological distinctions between Origen and Pelagius, she does affirm a trajectory from one to the other in their affirmation of human freedom. Pelagius is interested in championing human freedom, perhaps abstractly, but to particularly defend it against what he sees as Augustine's erring double election. The issue of infant baptism is at the forefront of the entire argument. Augustine, of course, would damn all unbaptized infants, for they have not received forgiveness for the sin they inherited. The only way Augustine can explain God's action in this regard is to declare that the unbaptized infants are not part of the predestined "elect". Julian states that it would be better not to believe in God than to believe that he exists and is not just. Additionally, Julian argues that God is not just if our condemnation is not based on our own freedom to choose good or evil. It is impossible for Julian to conceive of punishment for something "inherited", something beyond one's personal responsibility. While Clark has made clear the contrasts between Origen and Pelagius, their similar avowal of human freedom was

subverted while Augustine's determinism triumphed. His thought has been extremely persuasive in the Western church, but especially in the theology of Calvin and Luther. It is thus influential in the thought of Soren Kierkegaard, the Danish Lutheran, also known as the Melancholy Dane. As important as the question of how Augustine ever won the debate in the first place is, the reality is that most of Christianity accept the doctrine of original sin as a given—so much so that the only alternatives have been to accept it in whole or to reject it outright. Is there a middle way? Is there a theological option to affirm original sin as a condition of humankind without accepting what Augustine believed about its essence? In other words, can we confirm original sin but disrupt its singular definition as pride, mainly expressed as lust? The following is an attempt to light the way to exactly this particular via *media*.

### 2.2. Soren Kierkegaard on Anxiety

In *The Concept of Anxiety*, Soren Kierkegaard (SK) attempts to deal with a dilemma that had found little satisfaction in theology since the time of Augustine: how can a person be responsible for sins that he or she is destined, necessarily, to commit because of a precondition of corruption? Thus, the question moves from the guilt of having original sin to the effects of original sin's overpowering influence, which in turn, leads to the guilt of actual sins. Kierkegaard is interested in answering that dilemma without falling into Pelagianism and without denying human freedom in favor of strict double predestination. He succeeds in doing so by illuminating the relationship between the individual and the history of the human race, by distinguishing between the quantitative accumulation of original and a qualitative leap into personal sin, and by naming anxiety as the precondition into which each individual is born. He also believes every individual experiences his or her own fall.

Kierkegaard defines anxiety as the disposition that results from the synthesis of the self. The self is a synthesis of body and psyche, mediated by the spirit. In this synthesis, the self is aware of the freedom of potentiality; he or she is "able". "[A person] belongs to nature, but not to nature alone, for he is poised between nature and some other realm, and he is subject to imperatives that neither realm can explain of itself. He is material, yet spiritual; he is determined, yet free; he is derived similar to the rest of nature from what came before him, and yet, unlike anything else in nature, he alone is responsible for creating himself" (Price 1963). This causes anxiety. However, the human is really "anxious about nothing", and thus existentially anxious. In this scheme, anxiety is defined as a temper (to use Wesley's word) that results from this situation of living in the tension of the opposing poles. Anxiety arises in the space where the polarities are closely sensed: where the finite touches the infinite, where the temporal touches eternity, and where necessity touches freedom. This state of ambiguity gives rise to an anxiety woven into this created (not fallen) state. This anxiety was present in Adam and Eve (interestingly, more so in Eve since she excels in sensuality, or put differently, is closer to the earth, according to SK). They possessed "dreaming spirits" and were innocent by their ignorance of good and evil. (This is an important point for SK in his argument against the abuses found in the traditional understanding of hereditary sin.) Good and evil, and the punishment of death could only be understood after the leap into sin, which then brings knowledge of sin and goodness. The leap into sin is assumed to be a way to ease intolerable anxiety.

Kierkegaard believes that actual awareness of this anxiety is directly related to the degree to which the individual is spirit. The "spiritless" will have a decreased or absent consciousness of anxiety. The qualitative leap into sin is not dependent, however, on the consciousness of anxiety. In a very real sense, there is no explicable transition from anxiety to sin; the sin that results from anxiety is always a qualitative leap and one that cannot be necessary (which moves him away from Augustine). Kierkegaard also deals with anxiety "about something", the anxiety that results from sin, and is particularly interested in exploring the relationship between anxiety and guilt. This will ultimately lead him to the relationship between anxiety and the leap of faith (where he goes so far as to say that faith

without anxiety is impossible). This leads him to the Atonement, and in his words, "anxiety is delivered" into the realm of "dogmatics". He will pick up much of this discussion in *The Sickness Unto Death*.

### 2.3. Soren Kierkegaard on Despair

As just described, for SK, anxiety is sin's precondition. Despair, on the other hand, is the immediate consequence of sin. Despair is the resulting sickness of the self. All individuals live in despair. At the very moment of sin itself, it is the will that is prominent for SK. The essence of sin is related to the will. When the will, anxious about "nothing", attempts to attain or remain in equilibrium without relating itself to the Power which constitutes the whole relation of the self, despair ensues. In other words, attempting to be a self apart from God results in sin and despair.

What makes SK most helpful for our discussion here is that he imagines that the individual will fall to one side of the polarity or the other. Aspects of the polarities line up on sides. Let us imagine falling toward the right or left. On the left, so to speak, we find Infinitude and Possibility that are attractive as a means of overcoming one's anxiety. This is an attempt to move away from the "earthly". When persons fall toward the right, also trying to deal with anxiety, they fall toward Finitude and Necessity and move further into the earthly. SK identifies the fall toward the left as defiance, as willing to be a self apart from the Power (God) who constitutes the self. The fall toward the right he calls weakness, as not willing to be a self at all.

Important for our purposes here, SK wondered in a footnote if the forms of despair were perhaps related to gender. While men tend to despair by being willing to be a self apart from God, seeking only transcendence, SK thought that perhaps the despair of not being willing to be a self at all best characterizes "female despair", as women fall toward losing a self. Again, despair can be manifested in two ways: there are two ways of failing to be a true self. There is the despair of trying to be a self by oneself, which SK names the "manly" form of despair. This would parallel Augustine's concept of pride. There is also the opposite despair, the despair of not being willing to be a self at all, which he names the "womanly" form of despair. Man attempts to overcome the anxiety of selfhood by forcing the poles of infinitude and possibility, moving toward a groundless transcendence. Woman, on the other hand, relinquishes herself to the poles of finitude and necessity. Woman, according to SK, gives herself away, thus losing her true self. He writes, "This form of despair is: despair at not willing to be oneself, or still lower, despair at not willing to be a self; or lowest of all, despair at willing to be another than [her]self" ([Kierkegaard 1954](#)). The man, in contrast, defiantly attempts to maintain himself independently and egotistically, despairingly determined to be himself apart from God. The woman attempts to be rid of herself by losing herself to another. "Defiance" and "weakness" are Kierkegaard's final labels for the masculine and feminine forms of despair, respectively. Although he primarily called the female form of despair weakness, he also used the word "devotion". This dyad and specifically the word "devotion" are easily borrowed: women who are overly "devoted" to human relationships suffer from female despair and do not will to be a self; in other words, they suffer from too little self-esteem, indeed too little self.

Working closely with the Danish text, Sylvia Walsh interprets a key passage in Kierkegaard: "In abandoning or throwing herself altogether into that which she devotes herself, woman tends to have a sense of self only in and through the object of her devotion. When the object is taken away, her-self is also lost. Her despair, consequently, lies in not willing to be herself, that is, in not having any separate or independent self-identity", ([Walsh 1987](#)).

Mary Daly and others appropriately ask how women can be blamed for the "sin" or failure to self-actualize when social and cultural conditions, namely oppression, prevent them from doing so? In Daly's early work, when she was still interested in traditional theological categories, she offered a shift in the definition of original sin. For Daly, a women's original sin is the "internalization of blame and guilt". She continues, "The phrase

'original sin' is then torn from its original semantic context. The new sense retains the connotation of an inherited defect. However, it is understood that the 'sin' is inherited through socialization processes. It is the inherited burden of being condemned. The fault should not be seen as existing primarily in victimized individuals, but rather in demonic power structures that induce individuals to internalize false identities". Daly believes that rather than reinforcing stereotypes by labeling sin "masculine" or "feminine", healing will come when members of both sexes move toward "androgynous being". She states, "for women, this means exorcism of the internalized patriarchal presence, which carries with it feelings of guilt, inferiority, and self-hatred", (Daly 1973; Leclerc and Peterson 2022). While Daly's advice for androgyny made sense in early feminism, the option of empowered and liberated womanhood, vis à vis as women, seems perhaps more satisfactory in the context of Christian and particularly Wesleyan paradigms.

*2.4. Wesley on Sin*

John Wesley had very few, if any, positive comments on Augustine. While he does write a treatise on original sin, we really see his hamartiology played out in his practical works. Wesley wrote extensively to women (over 800 letters). In his letters, it is easy to spot his preference for keeping "his" women single and his concern that husbands and children do not dissuade a woman's spiritual commitment; he defines this practically and theologically. Wesley's primary concern was that those under his spiritual direction were single in heart. However, interestingly, Wesley did not use the more typically coupled opposite, "double-mindedness", in conjunction with this theological tenet. Rather, he repetitiously chose to use his preferred alternative phraseology: "keep yourself from idols". This is especially evident in the 1770s and 1780s when Wesley was reaching his mature theological convictions.

One young woman wrote to Wesley about a particular conflict in her band; Wesley, in turn, determined that she had been far too affected by the participants in the situation. "O beware of setting up any idol in your heart!" (Wesley 1770). In a later letter, he extended his advice. "As your mind is tender and easily moved, you may readily fall into inordinate affection; if you do, that will quickly darken your soul" (Wesley 1772). To another young woman, Ann Taylor, he penned, "Your real temptation will be, especially while you are young, to seek happiness in some creature. It is well if you are not entangled already—if you do not already begin to think, 'Oh, how happy I should be if I were to spend my life with this or that person. Vain thought! Happiness is not in man; no, nor in any creature under heaven . . . . No. When you begin to know God as your God, then, and not before, you begin to be happy . . . . [Therefore], by almighty grace, keep yourself from idols" (Wesley 1787).

Wesley's own perception of emotional dependency was limited. Yet, an alternative to the Augustinian definition of sin as pride is present in aspects of Wesley's own (gender) theory and practical relationships with women. He uses the words idolatry and inordinate affection endlessly in his correspondence with women. This parallels easily with SK's concept of devotion. He is concerned that women shift their devotion from God to a creature—to a human relationship. In 1749, Wesley said quite matter-of-factly to his own sister:

> I believe the death of your children is a great instance of the goodness of God towards you. You have often mentioned to me how much of your time they took up! Now the time is restored to you, and you have nothing to do but serve our Lord without carefulness and without distraction till you are sanctified body, soul, and spirit. (Wesley 1742)

Martha Wesley Hall had lost nine out of ten children in infancy. Her husband also left her after twenty years of marriage, which seems to have pleased Wesley for similar reasons.

In 1743, John Wesley penned "Thoughts Upon Marriage and the Single Life", which was redrafted as "Thoughts on a Single Life" in 1765. Here, Wesley exhorts anyone "called"

to the single life to "know the advantages you enjoy!" After listing many such advantages, he writes:

> Above all, you are at liberty from the greatest of all entanglements; the loving one creature above all others. It is possible to do this without sin, without any impeachment of our love to God. However, how inconceivably difficult! to give God our whole heart, while a creature has so large a share of it! How much more easily may we do this, when the heart is, tenderly indeed, but equally attached to more than one; or, at least, without any great inequality! What angelic wisdom does it require to give enough of our affection, and not too much to so near a relation! (Wesley 1765)

Wesley also wrote a sermon entitled "Spiritual Idolatry", which he penned nearer the end of his life. It will be helpful to quote one passage at length:

> Undoubtedly it is the will of God that we should all love one another. It is his will that we should love our relations and our Christian brethren with a peculiar love; and those in particular, whom he has made particularly profitable to our souls. These we are commanded to "love fervently"; yet still "with a pure heart". However, is not this "impossible with man?" to retain the strength and tenderness of affection, and yet, without any stain to the soul, with unspotted purity? I do not mean only unspotted by lust. I know this is possible. I know a person may have an unutterable affection for another without any desire of this kind. However, is it without idolatry? Is it not loving the creature more than the Creator? . . . It cannot be denied, that [persons] ought to love one another tenderly: they are commanded so to do. However, they are neither commanded nor permitted to love one another idolatrously. Yet, how common is this! How frequently is a husband, a wife, a child, put in the place of God. How many that are accounted good Christians fix their affections on each other, so as to leave no place for God! They seek their happiness in the creature, not in the Creator. One may truly say to the other, I view thee, lord and end of my desires. That is, "I desire nothing more but thee! Thou art the thing that I long for! All my desire is unto thee, and unto the remembrance of thy name". Now, if this is not flat idolatry, I cannot tell what is. (Wesley 1781)

*2.5. John Wesley on Full Salvation*

It is hopefully evident that John Wesley had a great practical interest in keeping Methodist women as employed in the Kingdom of God as possible. He was invested in his female lay preachers, in his female society leaders, and in the empowerment of all women. It would be possible to excavate his letters to women in order to find not only the sin of idolatry but to explicate a countering soteriology. We turn, however, to the more generalized theological insights of John Wesley as a way to address the concerns of "women's sin" and "women's salvation" found in the musings of SK. We will, therefore, look at his doctrine of prevenient grace in detail and his understanding of salvation and sanctification briefly—which all point toward a soteriology that envisions full salvation as therapeutic healing.

Long seen as foundational to Wesley's entire theology, the concept of prevenient grace has largely been assumed rather than specifically emphasized. Its significance can be seen when overlaid on the West's pervasive concept of depravity. Wesley's understanding of prevenient grace helps us envision the potentiality for something different. Fundamental to any doctrine of original sin is the belief that humanity is unable to rectify the situation in which it finds itself. God is the only means by which a fractured self (an egotistical self or a lack of self) is given any potentiality for change. However, by what means are we saved? How are we rescued out of absolute depravity, the West asks? Augustine and the Reformed tradition's emphasis on such depravity necessarily led them to the doctrine of predestination. Only God can save. Salvation must come through election because a

person is utterly incapable of doing anything—even enacting faith—in order to participate in God's salvation. It is in his arguments with Pelagius that Augustine is finally led to this extreme position.

Void of an understanding of prevenient grace, we find no way of explaining how the steps of salvation are possible except as an act of God alone (unconditional election) or as an act of the human will alone (thus denying original sin). If we want to affirm some form of original sin without falling into determinism, prevenient grace is undeniably crucial. If it is the presence of God that effectively vanquishes the power of sin, is it we who must find our way into that presence? Or is there a movement toward us, even in our despair, that does not override our will? It is here that the power of the concept (and certainly the reality) of prevenient grace solves the problem of deciding between predestination and Pelagianism, or in deciding to accept original sin as Augustine conceived it, or to throw it out altogether.

George Prince raises the question of whether SK sees the potentiality of selfhood in our despair or whether our original sin has obliterated this possibility. "[Our] basic condition makes ultimate self-improvement impossible. This is the force of the Greek phrase from Aristotle meaning 'in terms of possibility', e.g., as the oak-tree is in the acorn, the chicken in the egg . . . " According to Prince, for SK, "The self we desire to become is not even potentially present, and all efforts to develop it from the existing basis are futile until the self is in 'equilibrium'. Then, and only then, does the self exist 'in terms of possibility, and presents in itself an ideal basis for satisfactory development . . . 'Becoming' can now take place" (Price 1963). Prince places SK in a Reformed mode and explicitly states that there is no *possibility* of selfhood when we are in despair. The problem with this interpretation is that there is no explanation of how the self finds itself "in equilibrium" apart from a monergistic act of God. Yet, the will is very active in SK's theological anthropology. The will leaps into sin, and the will leaps into faith, but not in the total freedom of Pelagius. This raises for us an interesting question: does SK have a doctrine of prevenient grace?

For Wesley, the presence of God, through the gracious activity of the Holy Spirit, enables the will not to save itself but to move toward God, who is moving toward us. This is different from Pelagius' conceptualization of the activity of the will. We, meaning all humans, have free will because we have been graced by the very presence of God. We have graced free will. In the Reformers, true humanity is not even potentially present in those who are in despair, who have leaped into sin (in some sense, inevitably, if not necessarily). However, rather than waiting for salvation for the potentiality to be restored, Wesleyan theology opens up the possibility of true human potential in those who have yet to find "equilibrium" and salvation. Prevenient grace, then, reveals that the image of God remains in us, although it is distorted. Our very nature changes in terms of its potentiality immediately from the moment of birth (life) because we are not only affected by original sin—such sin is "counteracted" by prevenient grace. Our potentiality is restored before we are in equilibrium. If SK defines equilibrium itself as living in the tension between the poles of possibility and necessity, he certainly stands (with Wesley) in the tension between theologians affirming predestination and theologians who deny original sin.

How does this relate to the despair of not being willing to be a self at all? How is this applicable to women in despair through "inordinate affection" or "devotion"? How is this relevant to melancholy dames? Put simply, prevenient grace is the first movement of the healing of the disease of original sin. More specifically, prevenient grace offers the potentiality of self-hood for those whose will is so wounded that the very act of willing (or birthing) a healthy individuality is beyond them. Prevenient grace restores the very possibility of becoming a self, aids the wounded will toward wholeness, and enables a leap of faith.

Faith is imperative for Wesley and Kierkegaard. For both of them, faith is synergistic, a divine act and a participatory response. For those who will to be a self apart from the power of God, faith is a type of relinquishing of radical independence that aspires toward an impossible transcendence. Kierkegaard calls this infinite resignation. For those who do not

will to be a self or who will to be another('s) self, faith is embracing one's full potentiality by the empowerment of God and by God-directed devotion. Both infinite resignation and empowerment are central to Wesley's understanding of sanctification, which one might call a journey toward true personhood. What we find in Wesley and in Kierkegaard is the call for women to will to be a self, not devoted to a creature which leads to idolatry, but fully devoted to the God who begins to actualize her full range of possibility as the means of equilibrium (constituting selfhood).

Wesley can hardly imagine how this devotion to God can be maintained in a married state. Kierkegaard hints in the same direction. "Woman, with genuine womanliness, plunges herself into that to which she devotes herself . . . . It must be remembered, however, that we are not speaking here of devotion to God . . . . In relationship to God, where such a distinction as man/woman vanishes, it is true of man as of woman that devotion is the self, and that by devotion the self is acquired. This is true equality for man and woman", (Kierkegaard 1954). Sanctifying faith believes that all things are possible with God.

While the full equality of women cannot be anticipated fully by Wesley and SK, within their respective theologies, we find visions and depictions of empowered holy female selves. Indeed, they hope for it; it is hope that overcomes despair. In Wesleyan theology in particular, full salvation is presumed in the process of sanctification, where "despair as the sickness unto death" or "sin as disease" is healed and overcome. Wesley's soteriology is thoroughly therapeutic. Whereas soteriology in reformed theologies tends to be forensic, Wesleyan theology offers a picture of realignment, renewal, and new being for all genders.

### 3. Conclusions

The image of God in humans has been interpreted in many different ways through the centuries. One definition from a Wesleyan perspective is the capacity to love and be loved. We were created for loving relationships—with God, with ourselves, with other humans, and with the earth. Relationships of "inordinate affection" or improperly placed "devotion", on the other hand, result in a relational fracturing or a distortion of the imago Dei as God intends it to be. A common way to speak of this distortion, in general, is to speak of a "depravity" in humanity as the result of sin, usually imagined as the sin of pride, normed by Augustine and his Western followers. However, the distortion can also come, and this is key, *from being unable to be ones true self*, whether that be internally through a semblance of agency that does not will to be a self or externally under conditions of oppression and injustice where a person is not allowed to actualize selfhood. What is necessary when one is incapable to "save" themselves from this situation? A grace that goes before, a grace that enables a leap of faith, indeed, a prevenient grace. In this situation, the salvation that brings equilibrium can also be imagined as therapeutic healing in those who have been so victimized (at no fault of their own) that a *will to be* is beyond them. In other words, the wounded and diseased may participate in a grace that revives and renews their capacity to choose to be and then, consequentially, to love and be loved in healing and healthy ways. "Wesley was convinced that when the re-creative Spirit is at work real change occurs. Not only are we granted a new status in Christ through justification but God does not leave us where we were; God inaugurates a new creation, restoring the relation to which we are called to mirror God in the world. . . . what follows, the real change, is the beginning of new creaturehood, the telos toward which salvation is directed" (Runyon 1998).

As women, this reviving grace offers authentic, integrated personhood for those who have been fractured by sin or the sin of others. Foremost in this is the hope that comes from knowing ourselves as loved by God in truly transforming ways. God's healing grace comes in the form of spiritual empowerment as the self wills to be its true self in the act of faith. Such devotion delivers us from the darkness of despair that threatens our being. Additionally, such devotion opens us to a sanctified, wholehearted personhood that images the very nature of God.

**Funding:** This research received no external funding.

**Institutional Review Board Statement:** Not applicable.

**Informed Consent Statement:** Not applicable.

**Data Availability Statement:** Not applicable.

**Conflicts of Interest:** The author declares no conflict of interest.

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
