# Peer review of "“The Melancholy Dames”: Soren Kierkegaard’s Despairing Women and Wesley’s Empowering Cure"

_religions, doi:10.3390/rel14020144_

Round 1

Reviewer 1 Report

This brings insights for figuring out and imagining the true woman devoid of negative cultural and social and religious entanglements. conclusion can be improved based on kirkegaard idea of "leap" as a proposed antidote for despair.

leap factors for human human participation is wrought the work of work of salvation. For SK leap is comparable to Wesley's sanctification. Since Kirkegaardian philosophy is taken in the paper, SK's stages of life can be incorporated to strengthen  the struggles of being in becoming.

Authors first section can be simplified in terms of English.  

Author Response

Thank you for your feedback.

I will incorporate SK's leap into my section on sanctification.

And I will attend to your suggestion about his Stages.

I appreciate your insight.

Reviewer 2 Report

I believe the review of this article can be problematic. The line of reasoning goes through the apparently distant fields, and it is sometimes difficult to find and keep the main thread of the text. However, it is because the task of the article is challenging. The authors offer sophisticated theological reflection, which is hugely interesting. The text seems to be too wordy in some parts of the article, yet, it also is because of its challenging purpose. 

Nevertheless, there are some shortcomings and things that must be corrected. First of all, it is about the abstract. In this form, it appears merely as a part of the text; it does not present the purpose, methods and research questions. Second, the text should be revised, mainly in regard to punctuation

In summary, I highly rate the article and recommend it for publication.

Author Response

Thank you for your feedback.

I will go through the article and increase clarity.

Most importantly, I will rewrite the abstract.  I wrote the abstract after I was asked to submit one for the special issue on original sin, before I wrote the article.  I will make sure to address the issues you have raised.

I appreciate your insight.

Reviewer 3 Report

The essay deals admirably with the issues of freedom, sin, and grace with dexterity and insight, particularly in regard to the situation of women. In an impressive way it synthesizes material pertaining to Augustine, Wesley, Kierkegaard, and Mary Daly. For these reasons it should be published, but ideally after the following concern has been addressed.

However, I have one serious reservation. The implication that Kierkegaard had a deficient appreciation of prevenient grace is very misleading. In his upbuilding and Christian discourses he repeatedly writes about the need to rely upon God for the initiation of the Christian life and for its continued growth. Often he exclaims "Without God, an individual can do nothing." Particularly in his discourses for communion on Fridays he often concludes by reminding the reader that the communicant had done nothing at all to prepare for communion, but that God had been the agent in stirring up yearning for God, repentance, and every other spiritually relevant passion. I would like for the author to correct this lacuna in the interpretation of Kierkegaard. Also, there is no engagement with the contemporary literature on this issue in Kierkegaard. As a result, the depiction of Kierkegaard is rather one-dimensional.

Author Response

Thank you for your feedback.

Long ago, I did my comprehensive exams on Wesley; Original Sin; and the Patristic period.  I do not consider myself a scholar of SK.  So, thank you for your invaluable insight into SK and prevenient grace.  I will correct my observation accordingly.